# Effects of Mediterranean Diet or Low-Fat Diet on Blood Fatty Acids in Patients with Coronary Heart Disease. A Randomized Intervention Study

**DOI:** 10.3390/nu13072389

**Published:** 2021-07-13

**Authors:** Monica Gianna Giroli, José Pablo Werba, Patrizia Risé, Benedetta Porro, Angelo Sala, Manuela Amato, Elena Tremoli, Alice Bonomi, Fabrizio Veglia

**Affiliations:** 1Centro Cardiologico Monzino, IRCCS, Via Parea, 4, 20138 Milan, Italy; pablo.werba@ccfm.it (J.P.W.); benedetta.porro@cardiologicomonzino.it (B.P.); amatomanuela87@gmail.com (M.A.); etremoli@gvmnet.it (E.T.); alice.bonomi@cardiologicomonzino.it (A.B.); fabrizio.veglia@cardiologicomonzino.it (F.V.); 2Department of Pharmaceutical Sciences, University of Milan, Via Balzaretti, 9, 20133 Milan, Italy; patrizia.rise@unimi.it (P.R.); angelo.sala@unimi.it (A.S.); 3IRIB Consiglio Nazionale delle Ricerche, Via Ugo la Malfa, 153, 90146 Palermo, Italy; 4Maria Cecilia Hospital, Via Corriera, 1, 48033 Cotignola, Italy

**Keywords:** Mediterranean diet, low-fat diet, blood fatty acids, coronary heart disease, secondary prevention

## Abstract

The Mediterranean diet (MD) prevents cardiovascular disease by different putative mechanisms, including modifications in the blood fatty acid (FA) profile. Polytherapy for secondary cardiovascular prevention might mask the effect of MD on the FA profile. This study was aimed to assess whether MD, in comparison with a low-fat diet (LFD), favorably modifies the blood FA profile in patients with coronary heart disease (CHD) on polytherapy. One hundred and twenty patients with a recent history of coronary stenting, randomized to MD or to LFD, completed 3 months of this open-label dietary intervention study. Diet Mediterranean-ness was evaluated using the Mediterranean Diet Adherence Screener (MeDAS) score. Both diets significantly reduced saturated FA (*p* < 0.01). Putative favorable changes in total *n*-3 FA (*p* = 0.03) and eicosapentaenoic acid plus docosahexaenoic acid (EPA + DHA; *p* = 0.04) were significantly larger with MD than with LFD. At 3 months, in the whole cohort, the MeDAS score correlated inversely with palmitic acid (R = −0.21, *p* = 0.02), and with palmitoleic acid (R = −0.32, *p* = 0.007), and positively with total *n*-3 FA (R = 0.19, *p* = 0.03), EPA (R = 0.28, *p* = 0.002), and EPA + DHA (R = 0.21, *p* = 0.02). In CHD patients on polytherapy, both MD and LFD shift FA blood composition towards a healthier profile, with a more favorable effect of MD on omega−3 levels.

## 1. Introduction

Numerous observational studies [1,2,3,4] and one large interventional study in high-risk subjects [5] consistently supported a role of Mediterranean Diet (MD) in the primary prevention of cardiovascular disease (CVD). Clinical studies on the effect of MD on the residual risk of patients with overt CVD are currently ongoing [6,7].

The cardiovascular benefits associated with MD have been ascribed to different components of this dietary pattern such as fatty acids (FA) species, antioxidants, vitamins and phytochemicals contained in food. Putative mechanisms by which MD may improve cardiovascular health include favorable changes in the blood FA composition and in lipoprotein levels, protection from oxidative stress, inflammation and platelet aggregation as well as increased production of beneficial metabolites or reduced production of harmful metabolites by the intestinal microbiota [8,9,10]. Although the relative contribution of each of these mechanisms to the cardiovascular benefits is unknown, changes in the blood FA composition might play a significant role, as FA themselves modulate pathways implicated in atherothrombosis, such as oxidative stress and inflammation. Indeed, the blood FA composition was proposed as a non-traditional and potentially modifiable cardiovascular risk factor [11].

We previously reported that patients with coronary heart disease (CHD) adhere less to the MD and have a less favorable FA profile than healthy controls, with significantly higher total saturated fatty acids (SFA C16:0, C22:0, and C24:0), higher levels of several monounsaturated fatty acids (MUFA 16:1, 20:1, 22:1, and 24:1) and lower levels of polyunsaturated fatty acids (PUFA 20:3n−9, 18:2n−6, 20:3n−6, 18:3n−3, and 20:5n−3). Moreover, we also observed that CHD status was positively associated with total SFA and levels of 16:0 and 24:0, and negatively associated with levels of 16:1n−7, 20:3n−6, and 20:3n−9 [12]; other authors had reported similar observations [13].

Whether eating a MD may offset these differences is unknown. In fact, previous data show a positive effect of the MD on the blood FA profile of subjects without overt clinical manifestations of atherosclerosis [14,15]; however, to the best of our knowledge, no data is available on the effect of the MD on the blood FA profile in patients with overt CVD on-treatment with several drugs that may themselves [16] potentially mask or counterweigh the favorable effect of the nutritional intervention.

Therefore, this study was aimed at investigating the change of the whole blood fatty acid profile induced by a MD, in comparison with a control Low-Fat Diet (LFD), in patients recently subjected to percutaneous coronary revascularization on multiple cardiovascular drugs. We additionally evaluated whether the blood FA composition is related to the extent of dietary Mediterranean-ness reached with the nutritional intervention in the whole cohort regardless of group assignment.

## 2. Materials and Methods

The RISMeD study (Randomized Interventional Study on Mediterranean Diet) was a single center, controlled, open-label, laboratory-blinded, randomized intervention study carried out between 2015 and 2018. Briefly, the study involved the recruitment of 130 Caucasian patients of Italian origin, aged between 30 and 75 years, identified among those admitted to the Center for programmed percutaneous coronary revascularization. Patients were contacted during hospitalization, informed about the study and enquired about their willingness to take part. The patients were pre-screened to assess their eligibility. Exclusion criteria were a history of diabetes, food allergies or intolerances, body mass index (BMI) below 19 or above 33, intake of any drug or food supplement containing probiotics, *n*−3 FA or antioxidants (medications of other types in any number and dose were allowed) or any condition that, at the discretion of the researcher, might have hampered the patients’ adherence to the diet or to the schedule of visits. The patients were interrogated at prescreening about their usual eating habits by administrating the Mediterranean Diet Adherence Screener (MeDAS) questionnaire [17] and only those candidates with a MeDAS score at baseline ≤10 (out of a maximal score of 14) were included in the study to improve the chance of achieving measurable effects with the nutritional intervention.

Eligible candidates were enrolled about 2 months after the coronary revascularization procedure (T0). After signing the study informed consent, a complete clinical history was obtained (including detailed data on life-style habits and intake of medicines) and fasting peripheral venous blood was collected to determine FA composition in whole blood using gas chromatography, as described in Marangoni et al. [18]. Participants were randomly assigned (1:1) to a MD (*n* = 64) or to a standard LFD (control diet, *n* = 66), using a blocked randomization scheme (block size of six) (Figure 1). Patients randomized to the MD received a personalized dietary plan advice including on average: fish at least 3 times a week, legumes 2–3 times a week, raw or cooked vegetables 2 times a day, fruit 3 times a day, nuts 2–3 times a week, extra virgin olive oil 30–40 milliliters a day, not more than 150 g a week of red meat, red wine during meal: 1–2 glasses a day for men, 1 glass a day for women (only for subjects that usually drink wine, not for teetotalers). The consumption of cold cuts, sweets and soft drinks, butter, and full fat cheeses was discouraged. The control group received LFD—as recommended by cardiovascular prevention guidelines in force at the time of enrolment—which was high in vegetables, fruit and whole grains, and low in saturated fat (<10% of total caloric intake) and trans unsaturated fats [19]. The two dietary interventions mainly differed in the higher quantities of fish, legumes, and nuts in MD than in LFD.

In both groups, the diets were organized in seven days and personalized in terms of total calories to reduce weight in patients with BMI > 25 or to maintain weight otherwise. All the diets were developed with the software “Terapia Alimentare” of DS Medica Milan, Italy.

Patients carried out monthly visits (T1, T2 and T3). At each time, a nutritionist reinforced the individualized dietary recommendations. At T3, venous blood was collected for analysis of blood FA composition as at baseline, and the level of dietary Mediterranean-ness reached with the nutritional interventions was evaluated using the MeDAS score.

As the average baseline Italian’s dietary Mediterranean-ness is already at least moderate and the MD and LFD partly overlap in terms of food types, the score was administered to both groups to detect even subtle changes in dietary Mediterranean-ness.

The present study was conducted in accordance with the European Union’s Good Clinical Practice Standards and the Helsinki Declaration. The study was approved by the Review Board and the Ethics Committee of the hospital.

This study was registered in ClinicalTrials.gov: identifier no. NCT02578329; this paper describes the results regarding the primary endpoint of the study.

### Statistical Analysis

A total sample size of 120 subjects provided an estimated 80% statistical power to detect as significant (*p* < 0.05) a between group difference of at least 0.52 standard deviations; for instance, considering the variations reported in Mayneris-Perxachs, 2014 [15], the minimum detectable difference in palmitic acid variation was estimated as 1.33%.

Continuous variables were presented as mean ± standard deviation and were compared using the t-test for independent samples. Blood FA at T0 and T3 were compared using the paired Student T-Test. Categorical data were compared using the chi-squared test. Associations between variables were determined using Spearman’s rank correlation. A two-sided *p*-value less than 0.05 was required for statistical significance. All analyses were performed using the SAS statistical package V.9.4 (SAS Institute, Cary, NC, USA).

## 3. Results

The flow chart of the study is shown in Figure 1. At T0, 130 patients who met the inclusion criteria were randomized to MD (MD group, *n* = 64) or LFD (LFD group, *n* = 66). There were six dropouts in the MD group and four dropouts in the LFD group throughout the study. Thus, 120 patients completed the three months dietary intervention (58 MD, 62 LFD).

The baseline characteristics of study participants, stratified by group, are shown in Table 1. As expected for a sample of patients with CHD, there was a predominance of males. The mean age was 62.2 years. The vast majority was on treatment with several cardiovascular drugs. On average, participants were overweight and had a fairly well controlled blood pressure, LDL-cholesterol and glycaemia. The groups were well balanced, with the exception of significantly lower levels of total cholesterol and LDL-cholesterol in the LFD group (possibly in relationship with a significant higher intake of lipid-lowering drugs). There were no differences in terms of baseline eating habits or MeDAS score between the two groups.

Table 2 shows data on diet mediterranean-ness (MeDAS score) and levels of blood FA of patients, stratified by treatment arm, at baseline (T0), at three months of intervention (T3) as well as their changes after the intervention (T3-T0). The MeDAS score significantly augmented by 2.6 points (±2.2) and by 1.0 point (±1.8) in the MD and LFD groups, respec-tively (both *p* < 0.0001), though these changes were significantly larger in the MD group than in the LFD group (*p* < 0.0001).

Both diets significantly reduced whole blood SFA and specific SFA species (18:0 and 20:0). A reduction of palmitic acid (16:0) was observed with both diets, but reached statistical significance only with MD. However, none of these effects differed between diet groups.

Total MUFA and some MUFA species (oleic acid and 24:1) augmented whereas palmitoleic acid diminished with LFD but none of these changes was observed with MD. However, the changes did not differ between groups.

Total PUFA as well as total *n*−6 PUFA significantly increased in the MD group but not in the LFD group. However, these changes did not differ significantly between groups. The single *n*−6 PUFA differently influenced by the diets was arachidonic acid (AA), which augmented with MD but not with LFD.

Conversely, total *n*−3 PUFA tended to augment with MD and to diminish with LFD, with a statistically significant difference between groups. In line with these findings, eicosapentaenoic acid (EPA) plus docosahexaenoic acid (DHA) significantly increased in the MD group but did not change in the LFD group, an effect significantly different between groups. Finally, the ratio *n*−6/*n*−3 PUFA tended to decrease with MD and to increase with LFD, with a significant difference between groups. For the sake of completeness, the table describes changes induced by both diets on other minor individual FA.

Figure 2 shows correlations between dietary Mediterranean-ness achieved at T3 and blood FA species, in the whole cohort regardless of group assignment. MeDAS significantly and positively correlated with the relative concentrations of total *n*−3 PUFA, EPA, and EPA + DHA, and negatively correlated with those of the more prevalent SFA (palmitic acid), the monounsaturated palmitoleic acid, some minor long-chain *n*−6 PUFA (22:4*n*−6 and 22:5*n*−6) and the ratio *n*−6/*n*−3 PUFA. For completeness, the figure also describes significant correlations between MeDAS and other minor FA species, including a positive correlation with the SFA 20:0. Correlations between each of the 14-item of MeDAS score achieved at T3 and blood FA species are illustrated in Appendix A.

## 4. Discussion

This study shows that, in patients with CHD, diet Mediterranean-ness exerts some distinctive favorable effects on the blood FA profile beyond those that may be obtained with a “prudent” LFD. Though the magnitude of dietary-induced changes in the distribution of individual FA in blood is relatively small, it is in line with that observed in previous studies [15,20]. Both diets reduced total SFA, and particularly palmitic acid (16:0) (borderline change with LFD), which is the most abundant blood SFA and that which accounted for the major difference in terms of blood SFAs between patients with CHD and healthy subjects in case-control studies [12,13,21,22]. Noteworthily, we found that the higher the Mediterranean-ness of the diet adopted by the participants, the lower the blood palmitic acid achieved. Additionally, MeDAS positively correlated with the relative percentage of the very long chain SFA 20:0, which was associated with protection against CVD in some studies [23]. Therefore, in terms of blood SFAs, diet Mediterranean-ness is linked with putatively positive metabolic effects that may add to those obtained just with an LFD.

The role in CVD of total MUFA and oleic acid, which is the most abundant MUFA species in blood, remains highly controversial [24,25] and, therefore, the clinical relevance of dietary-induced changes in these FA species are highly speculative. In this study, the LFD produced a significant increase in total MUFA and variable changes in specific MUFA species, but these effects did not differ significantly between groups. Yet, levels of palmitoleic acid after three months of dieting were inversely correlated with MeDAS. This observation might be meaningful, as several pieces of evidence indicate that palmitoleic acid—which is downstream to palmitic acid in the pathway of de novo lipogenesis—is associated with several CV risk factors such as hepatic steatosis, high blood pressure, dyslipidemia, type 2 diabetes as well as with CVD [26].

Previous reports have shown that total PUFA as well as total *n*−6 PUFA are lower in patients with CHD than in healthy controls [12,13]. In this study, total PUFA and *n*−6 PUFA similarly increased with both dietary interventions. The observed increase in the relative concentration of AA with MD (but not with LFD) might theoretically be detrimental, as AA is the substrate of pro-inflammatory eicosanoids with atherogenic/thrombogenic effects [27]. However, the relationship between AA levels and risk of CHD is highly controversial, with direct [12], null [22,28] or even inverse associations [13,29,30].

The dietary interventions produced opposite changes in total *n*−3 PUFA and in EPA + DHA, which significantly increased with MD and did not change or even diminished with LFD. In addition, the achieved MeDAS score directly correlated with levels of total *n*−3 PUFA, EPA, and EPA + DHA, which are considered protective against CVD [31,32,33,34]. Although in the present study FA were determined in whole blood, previous studies show that EPA + DHA in whole blood highly correlates with EPA + DHA in erythrocytes [r(2) = 0.79], namely the omega−3 index [35], which has been proposed as a potential marker of cardiovascular health [36]. The mean level of EPA + DHA in whole blood achieved with MD was 3.94%, corresponding to an omega−3 Index of about 5% [35], which is above the values (<4%) formerly associated with the highest risk of death from CHD [37].

All in all, the most evident different changes induced by LFD and MD on blood FAs were on omega−3, which have a range of reputed vasculoprotective effects [38,39], possibly related with the association between omega−3 blood FA levels and cardiovascular health reported in several observational studies [31,33]. In the present study, blood omega−3 levels were significantly correlated, as expected, with the number of fish servings per week and with the intake of nuts, (i.e., with two out of the three main food items that differed quantitatively between LFD and MD). Both food items have been independently associated with cardiovascular protection [40,41]. On one hand, a recent pooled analysis of data from four cohort studies indicates that a minimal fish intake of 175 g weekly is associated with lower risk of major CVD and mortality, specifically among patients with prior CVD [42]. These data are in line with the favorable effects of fish intake on CV outcomes observed in the only interventional study carried out so far, to the best of our knowledge, in patients with overt CVD [43]. On the other hand, though the intake of nuts has been associated with a reduction in incident cardiovascular events, coronary events and mortality in prospective cohort studies of subjects free of CVD at baseline [44,45], intervention studies assessing the specific effect of nuts on cardiovascular outcomes have not been performed in either primary or secondary prevention.

The results of ongoing studies [6,7] will shed light about the effect of a MD, typically rich in fish and nuts, on the residual risk of patients with a history of cardiovascular events, as those included in the present investigation.

As far as we know, this is the first study to compare the effects of the MD vs. a control LFD on the blood FA profile in patients with overt CVD on-treatment with drugs that may potentially mask or counterweigh the benefits induced by the nutritional intervention.

Yet, this study has some limitations. First, it was carried out in Italians, a population with a traditional Mediterranean nutritional background. In fact, though we excluded a priori patients with a high MeDAS score, the mean baseline diet Mediterranean-ness of participants was actually not low but average (MeDAS about 7). As a consequence, the shift in dietary Mediterranean-ness reached with the MD was smaller than what might have been expected in a patient cohort with a usual Western diet. The two dietary interventions compared in this investigation are alike in several characteristics, and this may have led to the observed increase of the mean MeDAS score also in the LFD group (Table 2). Moreover, in relation to each patient adherence and preferences, the change in diet Mediterranean-ness of individual participants was variable regardless group assignment. As a result, there was some overlap of MeDAS values between groups (data not shown), reducing the chance to fully reveal the impact of diet Mediterranean-ness in improving the blood FA profile.

Second, circulating FA were determined using a rapid analytical method in whole blood, which represents a mixed FA pool that may differ from those measured in plasma, lipoproteins or red blood cells, traditionally used in FA research [21,30]; however, beyond its practicality, blood FA represents a balanced proportion of all the circulating pools [46] and has been adopted in several studies [13,47,48,49].

In conclusion, the results of the present short-term randomized dietary intervention study show that MD and LFD produce some equivalent “heart-friendly” changes in blood FA, whereas MD as well as the Mediterranean-ness of any of the dietary interventions may add distinctive beneficial effects, favoring a shift of the blood FA composition towards a healthier profile in patients with overt CHD on multiple cardiovascular drugs. These changes add to other beneficial effects of MD on intermediate surrogates of cardiovascular health, pending the results of ongoing secondary prevention studies with cardiovascular endpoints.

## Figures and Tables

**Figure 1 nutrients-13-02389-f001:**
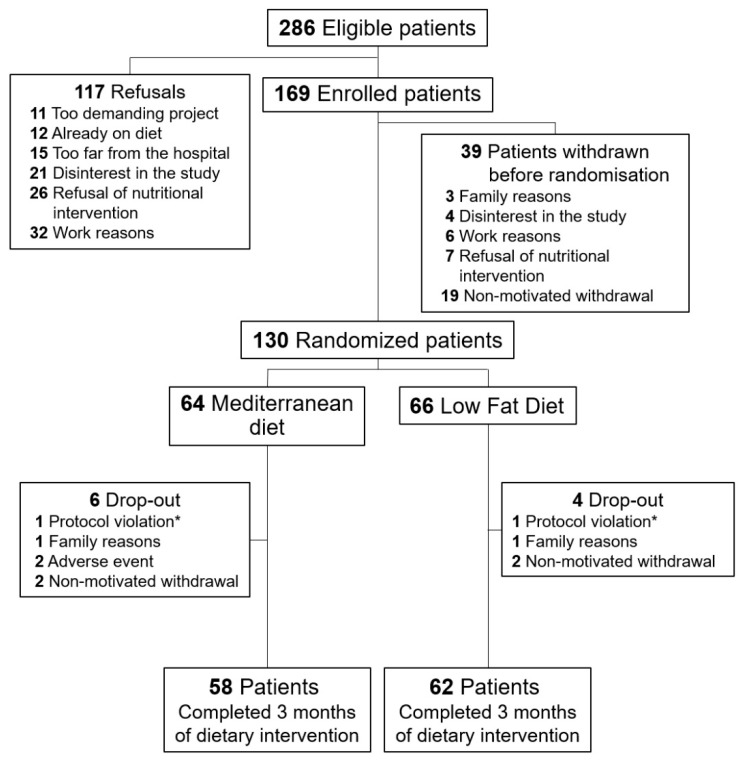
Recruitment chart. * Unnoticed intake of supplements containing omega−3 fatty acids.

**Figure 2 nutrients-13-02389-f002:**
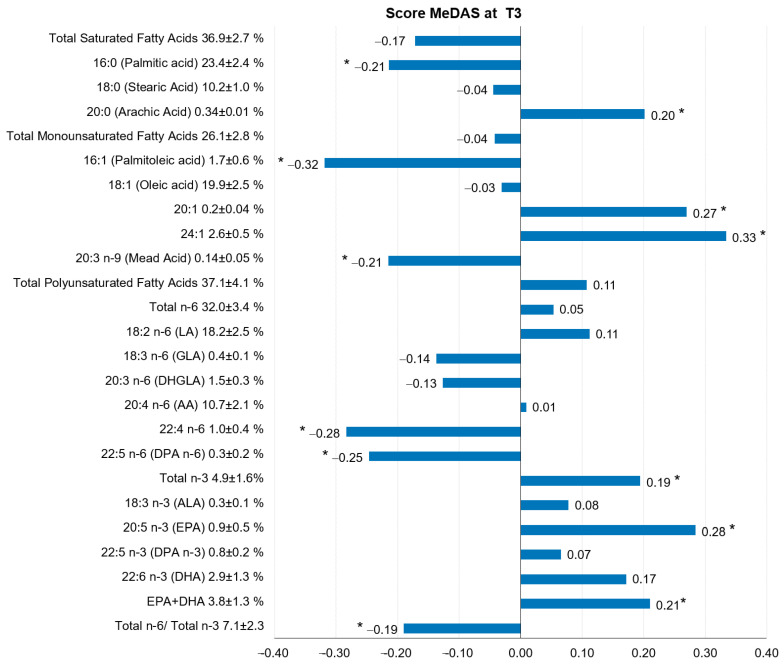
Correlations between the blood fatty acids profile and dietary Mediterranean-ness (MeDAS score) after three months of nutritional intervention (*n* = 120). MeDAS score: Mediterranean Diet Adherence Screener score; LA: linoleic acid; GLA: γ-linolenic acid; DHGLA: diomo-γ-linolenic acid; AA: arachidonic acid; DPA: docosapentaenoic acid; ALA: α-linolenic acid; EPA: eicosapentaenoic acid; DHA: docosahexaenoic acid. * *p*-value < 0.05.

**Table 1 nutrients-13-02389-t001:** Baseline population characteristics.

	ALL	MD	LFD	*p*-Value
	*N* = 130	*n* = 64	*n* = 66	
Men (*n*%)	113 (86.9%)	54 (84.4%)	59 (89.4%)	0.40
Age (years)	62.2 ± 9.5	63.0 ± 6.8	61.5 ± 11.6	0.35
Height (m)	1.70 ± 0.09	1.69 ± 0.09	1.70 ± 0.08	0.77
Body Weight (Kg)	79.2 ± 12.5	77.6 ± 12.2	80.8 ± 12.6	0.18
BMI (Kg/m^2^)	27.47 ± 3.46	27.1 ± 3.3	27.8 ± 3.6	0.31
Waist circumference (cm)	98.0 ± 9.8	97.0 ± 9.8	99.0 ± 9.8	0.33
Smoking (*n*%)				0.26
former smoker	77 (59.2%)	37 (57.8%)	40 (60.6%)	
current smoker	15 (11.5%)	5 (7.8%)	10 (15.2%)	
non-smoker	38 (29.2%)	22 (34.4%)	16 (24.2%)	
Physical activity (*n*%)				0.47
intense	29 (22.3%)	14 (21.9%)	15 (22.7%)	
average	41 (31.5%)	18 (28.1%)	23 (34.9%)	
low	60 (46.2%)	32 (50.0%)	28 (42.4%)	
Total cholesterol (mg/dL)	167.6 ± 29.2	172.7 ± 29.1	162.6 ± 28.5	0.05
HDL-cholesterol (mg/dL)	50.9 ± 13.0	49.8 ± 12.5	51.9 ± 13.5	0.45
LDL-cholesterol (mg/dL)	93.8 ± 25.5	99.3 ± 26.3	88.5 ± 23.6	0.02
Triglycerides (mg/dL)	114.4 ± 57.7	117.9 ± 63.2	111.1 ± 52.0	0.93
Fasting blood glucose (mg/dL)	101.9 ± 10.4	100.1 ± 9.1	103.7 ± 11.3	0.14
SBP (mmHg)	132.4 ± 18.0	132.8 ± 16.8	131.9 ± 19.3	0.63
DBP (mmHg)	79.4 ± 10.2	79.0 ± 10.9	79.9 ± 9.5	0.63
On lipid-lowering therapy (*n*%)	119 (91.5%)	55 (85.9%)	64 (97.0%)	0.02
On antihypertensive therapy (*n*%)	119 (91.5%)	60 (93.8%)	59 (89.4%)	0.37
On antiplatelet therapy (*n*%)	123 (94.6%)	61 (95.3%)	62 (93.9%)	0.73
Total medicines (median [min; max])	5 [3;9]	5 [3; 9]	5 [3; 9]	0.94
MeDAS Score	7.25 ± 1.6	7.36 ± 1.4	7.14 ± 1.8	0.47

Continuous variables are expressed as mean ± standard deviation; categorical data are presented as frequency and (percentage). MD: Mediterranean diet; LFD: low-fat diet; MeDAS score: Mediterranean Diet Adherence Screener score; SBP: systolic blood pressure; DBP: diastolic blood pressure.

**Table 2 nutrients-13-02389-t002:** Diet Mediterranean-ness and levels of fatty acids (%) of patients stratified per treatment arm at time T0, T3 and variation T3-T0.

	T0			T3			T3-T0				
	MD	LFD	*p*-Value	MD	LFD	*p*-Value	MD	*p*-Value	LFD	*p*-Value	*p*-Value
	*n* = 58	*n* = 62		*n* = 58	*n* = 62		*n* = 58	vs. T0	*n* = 62	vs. T0	Between Groups
MeDAS Score	7.3 ± 1.4	7.1 ± 1.8	0.55	9.9 ± 1.7	8.2 ± 1.7	<0.0001	2.6 ± 2.2	<0.0001	1.0 ± 1.8	<0.0001	<0.0001
Fatty acids levels (% of total fatty acids):											
Total Saturated Fatty Acids	38.5 ± 4.33	38.52 ± 4.07	0.77	36.39 ± 2.7	37.29 ± 2.72	0.03	−1.88 ± 3.98	0.001	−1.25 ± 3.66	0.009	0.4
16:0 (Palmitic Acid)	24.15 ± 3.21	24.35 ± 2.96	0.54	23 ± 2.45	23.71 ± 2.23	0.04	−0.89 ± 2.94	0.025	−0.65 ± 2.81	0.07	0.42
18:0 (Stearic Acid)	10.81 ± 1.64	10.77 ± 1.43	0.78	10.13 ± 0.96	10.32 ± 0.98	0.33	−0.71 ± 1.49	0.001	−0.45 ± 1.07	0.002	0.33
20:0 (Arachic Acid)	0.43 ± 0.27	0.43 ± 0.24	0.94	0.34 ± 0.09	0.33 ± 0.1	0.59	−0.08 ± 0.23	0.012	−0.1 ± 0.21	<0.001	0.26
Total Monounsaturated Fatty Acids	25.39 ± 3.65	24.81 ± 3.27	0.28	26.22 ± 3	25.89 ± 2.65	0.82	0.74 ± 3.4	0.1	1.08 ± 3.18	0.01	0.42
16:1 (Palmitoleic Acid)	1.76 ± 0.64	1.99 ± 0.81	0.09	1.59 ± 0.53	1.8 ± 0.68	0.08	−0.1 ± 0.48	0.1	−0.19 ± 0.58	0.012	0.9
18:1 (Oleic Acid)	19.3 ± 3.64	18.43 ± 3.24	0.12	20.13 ± 2.75	19.61 ± 2.31	0.51	0.7 ± 3.36	0.12	1.14 ± 3.31	0.009	0.34
20:1	0.2 ± 0.12	0.22 ± 0.19	0.84	0.19 ± 0.05	0.18 ± 0.03	0.66	0 ± 0.1	0.77	−0.04 ± 0.19	0.08	0.49
24:1	2.5 ± 0.56	2.46 ± 0.46	0.96	2.66 ± 0.47	2.62 ± 0.44	0.59	0.11 ± 0.58	0.17	0.17 ± 0.45	0.004	0.7
20:3 *n*−9 (Mead Acid)	0.21 ± 0.16	0.16 ± 0.11	0.03	0.14 ± 0.06	0.14 ± 0.04	0.82	−0.03 ± 0.11	0.05	−0.07 ± 0.14	0.001	0.12
Total Polyunsaturated Fatty Acids	36.11 ± 4.04	36.67 ± 4.78	0.38	37.39 ± 4.13	36.81 ± 4.11	0.31	1.14 ± 3.73	0.023	0.17 ± 4.14	0.75	0.1
Total *n*−6	31.06 ± 3.36	31.3 ± 4.23	0.56	32.15 ± 3.29	31.88 ± 3.51	0.58	0.95 ± 3.21	0.028	0.56 ± 3.56	0.22	0.4
18:2 *n*−6 (LA)	17.52 ± 2.99	17.1 ± 3.56	0.45	18.23 ± 2.15	18.09 ± 2.83	0.76	0.58 ± 2.8	0.12	0.91 ± 2.74	0.011	0.53
18:3 *n*−6 (GLA)	0.51 ± 0.29	0.54 ± 0.25	0.15	0.42 ± 0.15	0.42 ± 0.13	0.42	−0.07 ± 0.29	0.08	−0.12 ± 0.27	0.002	0.14
20:3 *n*−6 (DHGLA)	1.62 ± 0.35	1.54 ± 0.3	0.13	1.53 ± 0.3	1.43 ± 0.29	0.01	−0.11 ± 0.27	0.004	−0.11 ± 0.24	0.001	0.86
20:4 *n*−6 (AA)	10.13 ± 2.37	10.75 ± 2.44	0.13	10.73 ± 2.21	10.68 ± 1.99	0.69	0.6 ± 1.86	0.017	−0.01 ± 1.96	0.95	0.05
22:4 *n*−6	1.01 ± 0.42	1.08 ± 0.4	0.25	0.97 ± 0.35	1 ± 0.35	0.78	−0.06 ± 0.38	0.27	−0.07 ± 0.34	0.09	0.85
22:5 *n*−6 (DPA *n*−6)	0.27 ± 0.15	0.3 ± 0.16	0.25	0.27 ± 0.17	0.26 ± 0.12	0.54	0 ± 0.22	0.99	−0.04 ± 0.19	0.14	0.42
Total *n*−3	4.88 ± 1.85	5.17 ±1.83	0.24	5.11 ± 1.46	4.79 ± 1.54	0.2	0.22 ± 1.67	0.32	−0.32 ± 1.48	0.09	0.03
18:3 *n*−3 (ALA)	0.39 ± 0.24	0.39 ± 0.28	0.65	0.32 ± 0.11	0.34 ± 0.16	0.98	−0.03 ± 0.23	0.27	−0.05 ± 0.3	0.19	0.9
20:5 *n*−3 (EPA)	0.8 ± 0.47	0.88 ± 0.48	0.26	0.91 ± 0.5	0.83 ± 0.44	0.27	0.09 ± 0.56	0.22	−0.04 ± 0.41	0.417	0.07
22:5 *n*−3 (DPA *n*−3)	0.95 ± 0.53	1.02 ± 0.53	0.27	0.84 ± 0.24	0.81 ± 0.25	0.4	−0.11 ± 0.51	0.11	−0.2 ± 0.46	0.001	0.11
22:6 *n*−3 (DHA)	2.74 ± 1.06	2.88 ± 1.03	0.43	3.04 ± 0.87	2.82 ± 0.95	0.13	0.27 ± 0.89	0.023	−0.03 ± 0.92	0.77	0.12
EPA + DHA	3.58 ± 1.37	3.72 ± 1.42	0.38	3.94 ± 1.21	3.64 ± 1.30	0.18	0.36 ± 1.17	0.02	−0.07 ± 1.16	0.6	0.04
Total *n*−6/Total *n*−3	7.38 ± 3.01	7.01 ± 3	0.24	6.84 ± 2.16	7.35 ± 2.37	0.18	−0.62 ± 2.76	0.1	0.24 ± 2.36	0.42	0.04

Values are expressed as means ± standard deviation. Significant *p*-values are highlighted in bold. MD: Mediterranean diet; LFD: low-fat diet; MeDAS score: Mediterranean Diet Adherence Screener score; LA: linoleic acid; GLA: γ-linolenic acid; DHGLA: diomo-γ-linolenic acid; AA: arachidonic acid; DPA: docosapentaenoic acid; ALA: α-linolenic acid; EPA: eicosapentaenoic acid; DHA: docosahexaenoic acid.

## Data Availability

The raw dataset will be deposited in Zenodo repository.

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
