# Peer review of "Effects of Mediterranean Diet or Low-Fat Diet on Blood Fatty Acids in Patients with Coronary Heart Disease. A Randomized Intervention Study"

_nutrients, 2021, doi:10.3390/nu13072389_

Round 1

Reviewer 1 Report

Giroli et al. reported a randomized intervention study to assess the effects of MD on blood FA profile which would affect the incidence of CVD. The study design is appropriate and the results support the hypothesis that the MD changes the blood FA profile toward a favorable way compared to LFD. The reviewer requests the authors to answer one minor point as shown below. 

Minor comment:

1. Please clarify racial background of the studied cohort which might provide information whether the effects of MD differ among race or not. 

Reviewer 2 Report

Thank-you for your interesting manuscript presenting a randomized, controlled, parallel trial assessing the effect of personalized Mediterranean dietary advice compared to low fat dietary advice on the blood fatty acid profile in a group of 120 Italian adults with percutaneous coronary revascularization over a 3-month period. Findings showed the Mediterranean diet group proportionally increased PUFAs, decreased total saturated fatty acids (same as the control low fat diet group), and had no effect on MUFAs.

Comments/Suggestions:

Page 2, Lines 67-73. Suggest stating the primary and secondary objectives more clearly and directly as these seem to differ a little from the trial registration. This may also help provide context for the sample size determination (Page 3, Lines 126-129).

Page 2, Lines 81-89. Were there any specific inclusion/exclusion criteria, aside from the use of medications containing probiotics, n-3 FA, or antioxidants, regarding medication intake in terms of number of medications, type, adherence etc? Given a main aspect of the presented study is polytherapy this may be of interest to the reader and may also impact the findings.

Page 2, Line 88. Suggest providing context as to what the total MeDAS score is out of, for example “with a MeDAS score ≤ 10 out of a total possible score of 14 at baseline”.

Page 3, Line 97. Perhaps state that participants “received a personalize diet plan advice” as in this paragraph it is unclear as to whether participants would self-select their actual dietary intake or if food was provided, it isn´t until line 112 where it is noted the dietary intervention is recommendation based.

Page 3, Line 109. How was it determined who received dietary advice to maintain compared to reduce weight? Was this equal across the 2 study groups?

Page 4, Figure 1. Could the statement, “Takeover of exclusion criteria” please be clarified? Since inclusion/exclusion criteria is usually only intended to be assessed at baseline, and if something changes during the trial this should be considered and participants included, unless it was originally present at baseline but was missed in screening then it may be appropriate to exclude these individuals.

Page 4-5, Table 1. It would also be interesting to see the average total number and range (min number to max number) of medications participants in each group are taking (i.e. specifically addressing the polytherapy aspect).

Page 9, Lines 230-233. Given the Mediterranean dietary pattern is composed of components, such as olive oil and nuts, that are high in MUFAs in particular oleic acid, why do you think this was not reflected in the blood samples?

Thank-you for your time and consideration.
